# Adaptive Cholesky Gaussian Processes

## Abstract

We present a method to fit exact Gaussian process models to large datasets by considering only a subset of the data. Our approach is novel in that the size of the subset is selected on the fly during exact inference with little computational overhead. From an empirical observation that the log-marginal likelihood often exhibits a linear trend once a sufficient subset of a dataset has been observed, we conclude that many large datasets contain redundant information that only slightly affects the posterior. Based on this, we provide probabilistic bounds on the full model evidence that can identify such subsets. Remarkably, these bounds are largely composed of terms that appear in intermediate steps of the standard Cholesky decomposition, allowing us to modify the algorithm to adaptively stop the decomposition once enough data have been observed. Empirically, we show that our method can be directly plugged into well-known inference schemes to fit exact Gaussian process models to large datasets.

## 1 Introduction

It has been observed (Chalupka et al., 2013) that the random-subset-of-data approximation can be a hard-to-beat baseline for approximate Gaussian process inference. However, the question of how to choose the size of the subset is non-trivial to answer. Here we make an attempt. The key computational challenge in Gaussian process regression is to evaluate the log-marginal likelihood of the $N$ observed data points, which is known to have cubic complexity (Rasmussen & Williams, 2006). In order to arrive at a computationally less expensive approximation of this log-marginal likelihood, we first empirically study its behavior as we increase the number of observations. Figure 1 show this progression for a variety of models. We elaborate on this figure in Section 3.1, but for now note that after a certain number of observations, determined by the model and the dataset, the log-marginal likelihood starts to progress with a linear trend. This suggest that we may leverage this near-linearity to estimate the log-marginal likelihood of the full dataset after having seen only a subset of the data. However, as the point-of-linearity differs between models, this must be estimated on-the-fly to keep computations tractable.

In this paper, we approach the problem from a (frequentist) probabilistic numerics perspective (Hennig et al., 2015). By treating the dataset as a collection of independent and identically distributed random variables, a common assumption in the frequentist literature, we provide expected upper and lower bounds on the log-marginal likelihood, which become tight when the above-mentioned linear trend arises. We provide a particularly efficient algorithm for computing the bounds that leverage the intermediate computations performed by the Cholesky decomposition that is commonly used for evaluating the log-marginal likelihood. The bounds are therefore practically free to evaluate. We further show that these bounds allow us to predict *when* the linear trend determines the full-data log-marginal likelihood, such that we can phrase an *optimal stopping problem* to determine a suitable subset of the data for a particular model. We refer to our method as *Adaptive Cholesky Gaussian Process* (ACGP). Our approach has a complexity of $\mathcal{O}(M^3)$, where $M$ is the processed subset-size,

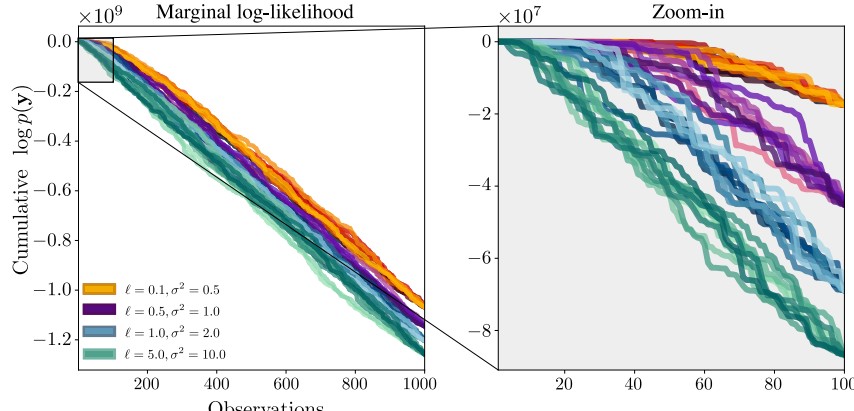

Figure 1: The figure shows total log-marginal likelihood as a function of the size of the training set of different permutations of a simple, synthetic dataset. The different colors correspond to different Gaussian process models, using the squared exponential kernel with length scale $\ell$ and amplitude $\sigma^2$. It can be seen that the log-marginal likelihood exhibits a linear trend after sufficiently many observations have been processed.

inducing an overhead of $\mathcal{O}(M)$ to the Cholesky. The main difference to previous work is that our algorithm does *not necessarily* look at the whole dataset, which makes it particularly useful in settings where even $\mathcal{O}(N)$ operations are intractable. When a dataset contains a large amount of redundant data, ACGP allows the inference procedure to stop early, saving precious compute—especially when the kernel function is expensive to evaluate.

## 2   Background

We use a PYTHON-inspired index notation, abbreviating for example $[y_1, \ldots, y_{n-1}]^\top$ as $\boldsymbol{y}_{:n}$; observe that the indexing starts at 1. With Diag we define the operator that sets all off-diagonal entries of a matrix to 0.

### 2.1   Gaussian Process Regression

We start by briefly reviewing Gaussian process (GP) models and how they are trained (see Rasmussen & Williams (2006, Chapter 2)). We consider the training dataset $\mathcal{D} = \{\boldsymbol{x}_n, y_n\}_{n=1}^N$ with inputs $\boldsymbol{x}_n \in \mathbb{R}^p$ and outputs $y_n \in \mathbb{R}$. The inputs are collected in the matrix $\boldsymbol{X} = [\boldsymbol{x}_1, \boldsymbol{x}_2, \ldots, \boldsymbol{x}_N]^\top \in \mathbb{R}^{N \times p}$. A GP $f \sim \mathcal{GP}(m(\boldsymbol{x}), k(\boldsymbol{x}, \boldsymbol{x}'))$ is a collection of random variables defined in terms of a mean function, $m(\boldsymbol{x})$, and a covariance function or *kernel*, $k(\boldsymbol{x}, \boldsymbol{x}') = \mathrm{cov}(f(\boldsymbol{x}), f(\boldsymbol{x}'))$, such that any finite amount of random variables has a Gaussian distribution. Hence, the prior over $\boldsymbol{f} := f(\boldsymbol{X})$ is $\mathcal{N}(\boldsymbol{f}; m(\boldsymbol{X}), \boldsymbol{K}_{\mathrm{ff}})$, where we have used the shorthand notation $\boldsymbol{K}_{\mathrm{ff}} = k(\boldsymbol{X}, \boldsymbol{X})$. We will consider the observations $\boldsymbol{y}$ as being noise-corrupted versions of the function values $\boldsymbol{f}$, and we shall parameterize this corruption through the likelihood function $p(\boldsymbol{y} \,|\, \boldsymbol{f})$, which for regression tasks is typically assumed to be Gaussian, $p(\boldsymbol{y} \,|\, \boldsymbol{f}) = \mathcal{N}(\boldsymbol{f}, \sigma^2 \boldsymbol{I})$. For such a model, the posterior over test inputs $\boldsymbol{X}_*$ can be computed in closed-form: $p(\boldsymbol{f}_* \,|\, \boldsymbol{y}) = \mathcal{N}(\boldsymbol{m}_*, \boldsymbol{S}_*)$, where

$$\boldsymbol{m}_* = k(\boldsymbol{X}_*, \boldsymbol{X})\boldsymbol{K}^{-1}\boldsymbol{y} \qquad \text{and} \qquad \boldsymbol{S}_* = k(\boldsymbol{X}_*, \boldsymbol{X}_*) - k(\boldsymbol{X}_*, \boldsymbol{X})\boldsymbol{K}^{-1}k(\boldsymbol{X}, \boldsymbol{X}_*).$$

with $\boldsymbol{K} := \boldsymbol{K}_{\mathrm{ff}} + \sigma^2 \boldsymbol{I}$. By marginalizing over the function values of the likelihood distribution, we obtain the marginal likelihood, $p(\boldsymbol{y}) = \int p(\boldsymbol{y} \,|\, \boldsymbol{f})p(\boldsymbol{f})d\boldsymbol{f}$, the de facto metric for comparing the performance of models in the Bayesian framework. While this integral is not tractable in general, it does have a closed-form solution for Gaussian process regression. Given the GP prior, $p(\boldsymbol{f}) = \mathcal{N}(\boldsymbol{0}, \boldsymbol{K}_{\mathrm{ff}})$, and the Gaussian likelihood, the log-marginal likelihood distribution can be found to be

$$\log p(\boldsymbol{y}) = -\frac{1}{2}\log\det[\boldsymbol{K}] - \frac{1}{2}\boldsymbol{y}^\top\boldsymbol{K}^{-1}\boldsymbol{y} - \frac{N}{2}\log 2\pi\,. \tag{1}$$

## 2.2 Background on the Cholesky decomposition

Inverting covariance matrices such as $K$ is a slow and numerically unstable procedure. Therefore, in practice, one typically leverages the Cholesky decomposition of the covariance matrices to compute the inverses. The Cholesky decomposition of a symmetric and positive definite matrix $K$ is the unique, lower[1] triangular matrix $L$ such that $K = LL^\top$ (Golub & Van Loan, 2013, Theorem 4.2.7). The advantage of having such a decomposition is that inversion with triangular matrices amounts to Gaussian elimination. There are different options to compute $L$. The Cholesky of a $1 \times 1$ matrix is the square root of the scalar. For larger matrices,

$$\mathrm{chol}[K] = \begin{bmatrix} \mathrm{chol}[K_{:s,:s}] & \mathbf{0} \\ T & \mathrm{chol}\left[K_{s:,s:} - TT^\top\right] \end{bmatrix}, \tag{2}$$

where $T := K_{:s,s:}\mathrm{chol}[K_{:s,:s}]^{-\top}$ and $s$ is any integer between $1$ and the size of $K$. Hence, extending a given Cholesky to a larger matrix requires three steps:

1. solve the linear equation system $T$,

2. apply the downdate $K_{s:,s:} - TT^\top$ and

3. compute the Cholesky of the down-dated matrix.

An important observation is that $K_{s:,s:} - TT^\top$ is the posterior covariance matrix $S_* + \sigma^2 I$ when considering $X_{s:}$ as test points. We will make use of this observation in Section 3.5. The log-determinant of $K$ can be obtained from the Cholesky using $\log \det [K] = 2 \sum_{n=1}^{N} \log L_{nn}$. A similar recursive relationship exists between the quadratic form $y^\top K^{-1} y$ and $L^{-1} y$ (see appendix, Equation (22)).

## 2.3 Related work

Much work has gone into tractable approximations to the log-marginal likelihood. Arguably, the most popular approximation methods for GPs are inducing point methods (Quiñonero-Candela & Rasmussen, 2005; Snelson & Ghahramani, 2006; Titsias, 2009; Hensman et al., 2013, 2017; Shi et al., 2020; Artemev et al., 2021), where the dataset is approximated through a set of pseudo-data points (inducing points), summarizing information from nearby data. Other approaches involve building approximations to $K$ (Fine & Scheinberg, 2001; Harbrecht et al., 2012; Wilson & Nickisch, 2015; Rudi et al., 2017; Wang et al., 2019) or aggregating of distributed local approximations (Gal et al., 2014; Deisenroth & Ng, 2015). One may also consider separately the approximation of the quadratic form via linear solvers such as conjugate gradients (Hestenes & Stiefel, 1952; Cutajar et al., 2016) and the approximation of the log-determinant (Fitzsimons et al., 2017a,b; Dong et al., 2017). Another line of research is scaling the hardware (Nguyen et al., 2019).

All above referenced approaches have computational complexity at least $\mathcal{O}(N)$ (with the exception of Hensman et al. (2013) since it uses mini-batching). However, the size of a dataset is seldom a particularly chosen value but rather the ad-hoc end of the sampling procedure. The dependence on the dataset size implies that more data requires more computational budget even though more data might not be helpful. This is the main motivation for our work: to derive an approximation algorithm where computational complexity does not depend on redundant data.

The work closest in spirit to the present paper is by Artemev et al. (2021), who also propose lower and upper bounds on quadratic form and log-determinant. There are a number of differences, however. Their bound relies on the method of conjugate gradients where we work directly with the Cholesky decomposition. Furthermore, while their bounds are deterministic, ours are probabilistic, which can make them tighter in certain cases, as they do not need to hold for all worst-case scenarios. This is also the main difference to the work of Hensman et al. (2013). Their bounds allow for mini-batching, but these are inherently deterministic when applied with full batch size.

---

[1]Equivalently, one can define $L$ to be upper triangular such that $K = L^\top L$.

## 3 Methodology

In the following, we will sketch our method. Our main goal is to convey the idea and intuition. To this end, we use suggestive notation. We refer the reader to the appendix for a more thorough and formal treatment.

### 3.1 Intuition on the linear extrapolation

The marginal likelihood is typically presented as a joint distribution, but, one can also view it from a cumulative perspective as the sum of log-conditionals:

$$\log p(\boldsymbol{y}) = \sum_{n=1}^{N} \log p(y_n \mid \boldsymbol{y}_{:n}) . \tag{3}$$

With this equation in hand, the phenomena in Figure 1 becomes much clearer. The figure shows the value of Equation (3) for an increasing number of observations $n$. When the plot exhibits a linear trend, it is because the summands $\log p(y_n \mid \boldsymbol{y}_{:n})$ become approximately constant, implying that the model is not gaining additional knowledge. In other words, new outputs are conditionally independent given the output observations seen so far.

The key problem addressed in this paper is how to estimate the full marginal likelihood, $p(\boldsymbol{y})$, from only a subset of $M$ observations. The cumulative view of the log-marginal likelihood in Equation (3) is our starting point. In particular, we will provide bounds, which are functions of seen observations, on the estimate of the full marginal likelihood. These bounds will allow us to decide, on the fly, when we have seen enough observations to accurately estimate the full marginal likelihood.

### 3.2 Stopping strategy

Suppose that we have processed $M$ data points with $N - M$ data points yet to be seen. We can then decompose Equation (3) into a sum of terms which have already been computed and a remaining sum

$$\log p(\boldsymbol{y}) = \underbrace{\sum_{n=1}^{M} \log p(y_n \mid \boldsymbol{y}_{:n})}_{\mathcal{A}:\ \text{processed}} + \underbrace{\sum_{n=M+1}^{N} \log p(y_n \mid \boldsymbol{y}_{:n})}_{\mathcal{B}:\ \text{remaining}} .$$

Recall that we consider the $\boldsymbol{x}_i, y_i$ as independent and identically distributed random variables. Hence, we could estimate $\mathcal{B}$ as $(N - M)\mathcal{A}/M$. Yet this is estimator is biased, since $(\boldsymbol{x}_{M+1}, y_{M+1}), \ldots, (\boldsymbol{x}_N, y_N)$ interact non-linearly through the kernel function. Instead, we will derive unbiased lower and upper bounds, $\mathcal{L}$ and $\mathcal{U}$. To obtain unbiased estimates, we use the last-$m$ processed points, such that conditioned on the points up to $s := M - m$, $\log p(\boldsymbol{y})$ can be bounded from above and below:

$$\mathbb{E}[\mathcal{L} \mid \boldsymbol{X}_{:s}, \boldsymbol{y}_{:s}] \leq \mathcal{A} + \mathbb{E}[\mathcal{B} \mid \boldsymbol{X}_{:s}, \boldsymbol{y}_{:s}] \leq \mathbb{E}[\mathcal{U} \mid \boldsymbol{X}_{:s}, \boldsymbol{y}_{:s}],$$

and the observations from $s$ to $M$ can be used to estimate $\mathcal{L}$ and $\mathcal{U}$. We can then detect when the upper and lower bounds are sufficiently near each other, and stop computations early when the approximation is sufficiently good. More precisely, given a desired relative error $r$, we stop when

$$\frac{\mathcal{U} - \mathcal{L}}{2 \min(|\mathcal{U}|, |\mathcal{L}|)} < r \quad \text{and} \quad \text{sign}(\mathcal{U}) = \text{sign}(\mathcal{L}) . \tag{4}$$

If the bounds hold, then the estimator $(\mathcal{L} + \mathcal{U})/2$ achieves the desired relative error (Lemma 16 in appendix). This is in contrast to other approximations, where one specifies a computational budget, rather than a desired accuracy.

### 3.3 Bounds

From Equation (1), we see that the log-marginal likelihood decomposes into the log-determinant of the kernel matrix, a quadratic term, and a constant term. In the following we present upper and lower bounds for both the log-determinant ($\mathcal{U}_{\text{D}}$ and $\mathcal{L}_{\text{D}}$, respectively) and the quadratic term ($\mathcal{U}_{\text{Q}}$ and $\mathcal{L}_{\text{Q}}$).

We will need the posterior equations for the observations, that is $p(\boldsymbol{y}_n \mid \boldsymbol{y}_{:n})$, and we will need them as functions of test inputs $\boldsymbol{x}_*$ and $\boldsymbol{x}'_*$. To this end, define

$$\boldsymbol{m}_*^{(n)}(\boldsymbol{x}_*) := k(\boldsymbol{x}_*, \boldsymbol{X}_{:n})\boldsymbol{K}_{:n,:n}^{-1}\boldsymbol{y}_{:n} \qquad \text{and}$$

$$\boldsymbol{\Sigma}_*^{(n)}(\boldsymbol{x}_*, \boldsymbol{x}'_*) := k(\boldsymbol{x}_*, \boldsymbol{x}'_*) + \sigma^2 \delta_{\boldsymbol{x}_*, \boldsymbol{x}'_*} - k(\boldsymbol{x}_*, \boldsymbol{X}_{:n})\boldsymbol{K}_{:n,:n}^{-1}k(\boldsymbol{X}_{:n}, \boldsymbol{x}'_*)$$

such that $p(y_n \mid \boldsymbol{y}_{:n}) = \mathcal{N}(y_n; \boldsymbol{m}_*^{(n)}(\boldsymbol{x}_n), \boldsymbol{\Sigma}_*^{(n)}(\boldsymbol{x}_n, \boldsymbol{x}_n))$, which allows us to rewrite Equation (3) as

$$\log p(\boldsymbol{y}) \propto \sum_{n=1}^{N} \log \boldsymbol{\Sigma}_*^{(n-1)}(\boldsymbol{x}_n, \boldsymbol{x}_n) + \sum_{n=1}^{N} \frac{(y_n - \boldsymbol{m}_*^{(n-1)}(\boldsymbol{x}_n))^2}{\boldsymbol{\Sigma}_*^{(n-1)}(\boldsymbol{x}_n, \boldsymbol{x}_n)} . \tag{5}$$

This reveals that the log-determinant can be written as a sum of posterior variances and the quadratic form has an expression as normalized square errors. Other key ingredients for our bounds are estimates for average posterior variance and average covariance. Therefore define the shorthands

$$\boldsymbol{V} := \mathrm{Diag}\left[\boldsymbol{\Sigma}_*^{(s)}(\boldsymbol{X}_{s:M}, \boldsymbol{X}_{s:M})\right] \qquad \text{and} \qquad \boldsymbol{C} := \sum_{i=1}^{\frac{M}{2}} \boldsymbol{\Sigma}_*^{(s)}(\boldsymbol{x}_{s+2i}, \boldsymbol{x}_{s+2i-1})\boldsymbol{e}_{2i}\boldsymbol{e}_{2i}^{\top} ,$$

where $\boldsymbol{e}_j \in \mathbb{R}^m$ is the $j$-th standard basis vector. The matrix $\boldsymbol{V}$ is simply the diagonal of the posterior covariance matrix $\boldsymbol{\Sigma}_*$. The matrix $\boldsymbol{C}$ consists of every *second* entry of the first off-diagonal of $\boldsymbol{\Sigma}_*$. These elements are placed on the diagonal with every second element being 0. The reason for taking every second element is of theoretical nature, see Remark 2 in the appendix. In practice we use the full off-diagonal.

### 3.3.1 Bounds on the log-determinant

Both bounds, lower and upper, use that $\log \det [\boldsymbol{K}] = \log \det [\boldsymbol{K}_{:s,:s}] + \log \det \left[\boldsymbol{\Sigma}_*^{(s)}(\boldsymbol{X}_{s:}, \boldsymbol{X}_{s:})\right]$ which follows from the matrix-determinant lemma. The first term is available from the already processed datapoints. It is the second addend that needs to be estimated, which we approach from the perspective of Equation (5). It is well-established that, for a fixed input, more observations decrease the posterior variance, and this decrease cannot cross the threshold $\sigma^2$ (Rasmussen & Williams, 2006, Question 2.9.4). This remains true when taking the expectation over the input. Hence, the average of the posterior variances for inputs $\boldsymbol{X}_{s:M}$ is with high probability an overestimate of the average posterior variance for inputs with higher index. This motivates our upper bound on the log-determinant:

$$\mathcal{U}_{\mathrm{D}} = \log \det [\boldsymbol{K}_{:s,:s}] + (N - s)\mu_D \tag{6}$$

$$\mu_D := \frac{1}{m}\sum_{i=1}^{m} \log (\boldsymbol{V}_{ii}) \quad /\!/ \text{ average log posterior variance}$$

To arrive at the lower bound on the log-determinant, we need an argument about how fast the average posterior variance could decrease which is governed by the covariance between inputs. The variable $\rho_D$ measures the average covariance, and we show in Theorem 6 in the appendix that this overestimates the decrease per step with high probability. Since the decrease cannot exceed $\sigma^2$, we introduce $\psi_D$ to denote the step which would cross this threshold.

$$\mathcal{L}_{\mathrm{D}} = \log \det [\boldsymbol{K}_{:s,:s}] + (\psi_D - s)\left(\mu_D - \frac{\psi_D - s - 1}{2\sigma^4}\rho_D\right) + (N - \psi_D)\log \sigma^2. \tag{7}$$

$$\rho_D := \frac{2}{m}\sum_{i=1}^{m} \boldsymbol{C}_{2i,2i}^2 \quad /\!/ \text{ average square covariance}$$

$$\psi_D := \max(N, \lfloor s - 1 + {}^2\!/\!_{\rho_D}\left(\mu_D - \log \sigma^2\right)\rfloor) \quad /\!/ \text{ steps } \mu_D \text{ can decrease by } \rho \tag{8}$$

Both bounds collapse to the exact solution when $s = N$. The bounds are close when the average covariance between inputs, $\rho_D$, is small. This occurs for example when the average variance is close to $\sigma^2$ since the variance is an upper bound to the covariance. Another case where $\rho_D$ is small is when points are not correlated to begin with.

### 3.3.2 Bounds on the quadratic term

Denote with $\boldsymbol{r}_* := \boldsymbol{y}_{s:} - \boldsymbol{m}_*^{(s)}(\boldsymbol{X}_{s:})$ the prediction errors (the residuals), when considering the first $s$ points as training set and the remaining inputs as test set. Analogous to the bounds on the log-determinant, one can show with the matrix inversion lemma that $\boldsymbol{y}^\top \boldsymbol{K}^{-1} \boldsymbol{y} = \boldsymbol{y}_{:s}^\top \boldsymbol{K}_{:s,:s}^{-1} \boldsymbol{y}_{:s} + \boldsymbol{r}_*^\top (\boldsymbol{\Sigma}_*^{(s)}(\boldsymbol{X}_{s:}))^{-1} \boldsymbol{r}_*$. Again, the first term will turn out to be already computed. With a slight abuse of notation let $\boldsymbol{r}_* := \boldsymbol{y}_{s:M} - \boldsymbol{m}_*^{(s)}(\boldsymbol{X}_{s:M})$, that is, we consider only the first $m$ entries. Our lower bound arises from another well-known lower bound: $\boldsymbol{a}^\top \boldsymbol{A}^{-1} \boldsymbol{a} \geq 2\boldsymbol{a}^\top \boldsymbol{b} - \boldsymbol{b}^\top \boldsymbol{A} \boldsymbol{b}$ for all $\boldsymbol{b}$ (see for example Kim & Teh (2018); Artemev et al. (2021)). We choose $\boldsymbol{b} := \alpha \boldsymbol{1}$ where $\alpha$ is chosen to maximize the bound. The result, after some cancellations, is the following lower bound on the quadratic term:

$$\mathcal{L}_Q = \boldsymbol{y}_{:s}^\top \boldsymbol{K}_{:s,:s}^{-1} \boldsymbol{y}_{:s} + (N-s)\alpha \left(2\mu_Q - \alpha\rho_Q\right) \tag{9}$$

$$\mu_Q := \frac{1}{m} \boldsymbol{r}_*^\top \boldsymbol{r}_* \quad /\!\!/ \text{ average square error}$$

$$\rho_Q := \frac{1}{m} \boldsymbol{r}_*^\top \boldsymbol{r}_* + \frac{N-s-1}{m} \sum_{j=\frac{s+2}{2}}^{\frac{M}{2}} \boldsymbol{r}_{*,2j} \boldsymbol{r}_{*,2j-1} \boldsymbol{C}_{2j,2j}$$

The $\alpha$ maximizing above bound is $\mu_Q/\rho_Q^2$, which is the value we chose in our implementation. However, note that $\mathcal{L}_Q$ is an expected lower bound only if $\alpha$ depends on variables with index smaller than $s$.

Our upper bound arises from the element-wise perspective of Equation (5). We assume that the expected mean square error $(y_n - \boldsymbol{m}_*^{(n-1)}(\boldsymbol{x}_n))^2$ decreases with more observations. However, though mean square error and variance decrease, their expected ratio may increase or decrease depending on the choice of kernel, dataset and number of processed points. Using the average error calibration with a correction for the decreasing variance, we arrive at our upper bound on the quadratic term:

$$\mathcal{U}_Q = \boldsymbol{y}_{:s}^\top \boldsymbol{K}_{:s,:s}^{-1} \boldsymbol{y}_{:s} + (N-s)\left(\mu_Q' + \rho_Q'\right) \tag{10}$$

$$\mu_Q' := \frac{1}{m} \boldsymbol{r}_*^\top \boldsymbol{V}^{-1} \boldsymbol{r}_* \quad /\!\!/ \text{ average error calibration}$$

$$\rho_Q' := \frac{N-s-1}{m} \frac{1}{\sigma^4} \boldsymbol{r}_*^\top \boldsymbol{C} \boldsymbol{V}^{-1} \boldsymbol{C} \boldsymbol{r}_* \quad /\!\!/ \text{ average increase in error calibration}$$

In our implementation we use a slightly different upper bound. The estimate of the possible decrease of the variance uses the same technique as the lower bound for the log-determinant. Therefore we can define an analogue to Equation (8) determining the step when the variance estimate falls below $\sigma^2$. In our implementation, addends of the quadratic after this step are estimated by the more conservative $\sigma^{-2}\mu_Q$. Again, the bounds collapse to the true quantity when $s = N$. The bounds will give good estimates when the average covariance between inputs, represented by the matrix $\boldsymbol{C}$, is low or when the model can predict new data well, that is, when $\boldsymbol{r}_*$ is close to $0$.

### 3.4 Validity of bounds and stopping condition

For the upper bound on the quadratic form, we need to make a (technical) assumption. It expresses the intuition that the (expected) mean square error should not increase with more data—a model should not become worse as its training set increases.[2]

**Assumption 1.** *Assume that*

$$\mathbb{E}\left[f(\boldsymbol{x}, \boldsymbol{x}')(y_j - \boldsymbol{m}_*^{(j-1)}(\boldsymbol{x}))^2 \mid \boldsymbol{X}_{:s}, \boldsymbol{y}_{:s}\right] \leq \mathbb{E}\left[f(\boldsymbol{x}, \boldsymbol{x}')(y_j - \boldsymbol{m}_*^{(s)}(\boldsymbol{x}))^2 \mid \boldsymbol{X}_{:s}, \boldsymbol{y}_{:s}\right]$$

*for all $s \in \{1, \ldots, N\}$ and for all $s < j \leq N$, where $f(\boldsymbol{x}, \boldsymbol{x}')$ is either $\frac{1}{\boldsymbol{\Sigma}_*^{(s)}(\boldsymbol{x}, \boldsymbol{x})}$ or $\frac{\boldsymbol{\Sigma}_*^{(s)}(\boldsymbol{x}, \boldsymbol{x}')^2}{\sigma^4 \boldsymbol{\Sigma}_*^{(s)}(\boldsymbol{x}, \boldsymbol{x})}$.*

**Theorem 2.** *Assume that $(\boldsymbol{x}_1, y_1), \ldots, (\boldsymbol{x}_N, y_N)$ are independent and identically distributed, assume that Assumption 1 holds, and assume that $\alpha$ in the definition of $\mathcal{L}_Q$ depends only on $\boldsymbol{x}_1, y_1, \ldots \boldsymbol{x}_s, y_s$.*

---

[2]Empirically, we confirmed this assumption for all experiments considered in Section 4.1 and in Appendices B.3.1 to B.3.3.

For any $s \in \{1, \dots, N\}$, the bounds defined in Equations (6), (7), (9) and (10) hold in expectation:

$$\mathbb{E}[\mathcal{L}_D \mid \boldsymbol{X}_{:s}, \boldsymbol{y}_{:s}] \leq \mathbb{E}[\log \det [\boldsymbol{K}] \mid \boldsymbol{X}_{:s}, \boldsymbol{y}_{:s}] \leq \mathbb{E}[\mathcal{U}_D \mid \boldsymbol{X}_{:s}, \boldsymbol{y}_{:s}] \text{ and}$$

$$\mathbb{E}[\mathcal{L}_Q \mid \boldsymbol{X}_{:s}, \boldsymbol{y}_{:s}] \leq \mathbb{E}[\boldsymbol{y}^\top \boldsymbol{K}^{-1} \boldsymbol{y} \mid \boldsymbol{X}_{:s}, \boldsymbol{y}_{:s}] \leq \mathbb{E}[\mathcal{U}_Q \mid \boldsymbol{X}_{:s}, \boldsymbol{y}_{:s}] \,.$$

Proof and a proof sketch can be found in the appendix.

**Theorem 3.** *Let $r > 0$ be a desired relative error and set $\mathcal{U} := \mathcal{U}_D + \mathcal{U}_Q$ and $\mathcal{L} := \mathcal{L}_D + \mathcal{L}_Q$. If the stopping conditions hold, that is, $\mathrm{sign}(\mathcal{U}) = \mathrm{sign}(\mathcal{L})$ and Equation (4) is true, then $\log p(\boldsymbol{y})$ can be estimated from $(\mathcal{U} + \mathcal{L})/2$ such that, under the condition $\mathcal{L}_D \leq \log(\det [\boldsymbol{K}]) \leq \mathcal{U}_D$ and $\mathcal{L}_Q \leq \boldsymbol{y}^\top \boldsymbol{K}^{-1} \boldsymbol{y} \leq \mathcal{U}_Q$, the relative error is smaller than $r$, formally:*

$$|\log p(\boldsymbol{y}) - (\mathcal{U} + \mathcal{L})/2| \leq r |\log p(\boldsymbol{y})| \,. \tag{11}$$

The proof follows from Lemma 16 in the appendix.

Theorem 2 is a first step to obtain a probabilistic statement for Equation (11), that is, a statement of the form $\mathbb{P}\left( \left| \frac{\log p(\boldsymbol{y}) - \frac{1}{2}(\mathcal{U} + \epsilon_{\mathcal{U},\delta} + \mathcal{L} - \epsilon_{\mathcal{L},\delta})}{\log p(\boldsymbol{y})} \right| > r \right) \leq \delta$. Theoretically, we can obtain such a statement using standard concentration inequalities and a union bound over $s$. In practice, the error guarding constants $\epsilon$ would render the result trivial. A union bound can be avoided using Hoeffding's inequality for martingales (Fan et al., 2012). However, this requires to replace $s := M - m$ by a stopping time independent of $M$, which we regard as future work.

## 3.5 Practical implementation

The proposed bounds turn out to be surprisingly cheap to compute. If we set the block-size of the Cholesky decomposition to be $m$, the matrix $\boldsymbol{\Sigma}_*^{(s)}$ is exactly the downdated matrix in step 2 of the algorithm outlined in Section 2.2. Similarly, the expressions for the bounds on the quadratic form appear while solving the linear equation system $\boldsymbol{L}^{-1}\boldsymbol{y}$. A slight modification to the Cholesky algorithm is enough to compute these bounds on the fly during the decomposition with little overhead.

The stopping conditions can be checked before or after Step 3 of the Cholesky decomposition (Section 2.2). Here, we explore the former option since Step 3 is the bottleneck due to being less parallelizable than the other steps.

Note that the definition of the bounds does not involve variables $\boldsymbol{x}, y$ which have not been processed. This allows an on-the-fly construction of the kernel matrix, avoiding potentially expensive kernel function evaluations. Furthermore, it is *not* necessary to allocate $\mathcal{O}(N^2)$ memory in advance; a user can specify a maximal amount of processed datapoints, hoping that stopping occurs before hitting that limit. We provide the pseudo-code for this modified algorithm, our key algorithmic contribution, in the appendix. Additionally, we provide a PYTHON implementation of our modified Cholesky decomposition and scripts to replicate the experiments of this paper.[3]

# 4 Experiments

We now examine the bounds and stopping strategy for ACGP. When running experiments without GPU support, all linear algebra operations are substituted for direct calls to the OPENBLAS library (Wang et al., 2013), for efficient realization of *in-place* operations. To still benefit from automatic differentiation, we used PYTORCH (Paszke et al., 2019) with a custom backward function for $\log p(\boldsymbol{y})$ which wraps OPENBLAS. The details of our experimental setup can be found in Appendix A.

## 4.1 Bound quality

In this section we examine the bounds presented in Section 3 and compare them to those proposed by Artemev et al. (2021, Lemma 2 and Lemma 3) (CGLB). Specifically, for the determinant we compare to their $\mathcal{O}(N)$ upper bound (Artemev et al., 2021, Eq. 11) and their $\log(\det [\boldsymbol{Q}])$ as lower bound.

We set the number of inducing inputs $M$ for CGLB to 512, 1024, 2048, and 4096. For ACGP, we define $m := 40 \cdot 256 = 10\,240$ which is the number of cores times the default OPENBLAS block

---

[3]The code is available at the following repository: anonymized

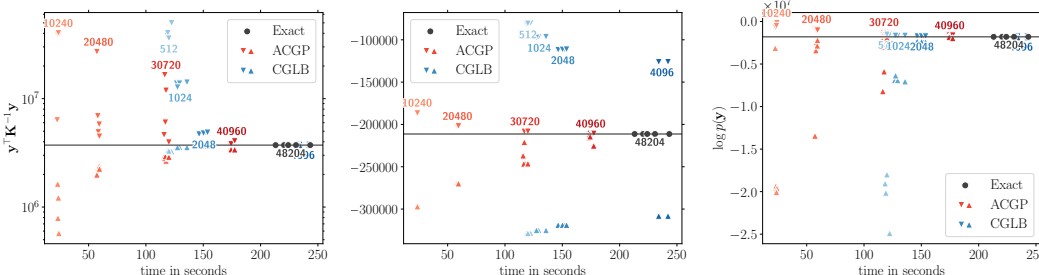

(a) Upper and lower bounds on the quadratic term.

(b) Upper and lower bounds on the log-determinant term.

(c) Upper and lower bounds on the full log-marginal likelihood.

Figure 2: Comparison of the upper and lower bounds for ACGP and CGLB on the `metro` dataset using the OU kernel with a length scale of $\log \ell = 0$ and the time it takes to compute them. The black line indicates the result obtained using exact GP regression with points above and below it marking the upper and lower bounds, respectively. The experiment was repeated five times with different seeds to illustrate the variability in the computation time, shown here as multiple points of the same color. For ACGP the number near the points shows $M$, the size of the used subset; for CGLB it is the number of inducing inputs. The color of the points reflects these numbers to help discern the size of the subset or number of inducing inputs from the repeated experiments.

size for our machines. We compare both methods using squared exponential kernel (SE) and the Ornstein-Uhlenbeck kernel (OU),

$$k_{\text{SE}}(\boldsymbol{x}, \boldsymbol{z}) := \theta \exp\left( -\frac{\|\boldsymbol{x} - \boldsymbol{z}\|^2}{2\ell^2} \right), \qquad k_{\text{OU}}(\boldsymbol{x}, \boldsymbol{z}) := \theta \exp\left( -\frac{\|\boldsymbol{x} - \boldsymbol{z}\|}{\ell} \right),$$

where we fix $\sigma^2 := 10^{-3}$ and $\theta := 1$, and we vary $\ell$ as $\log \ell \in \{-1, 0, 1, 2\}$. As benchmarking datasets we use the two datasets consisting of more than $20\,000$ instances used by Artemev et al. (2021): `kin40k` and `protein`. We further consider two additional datasets from the UCI repository (Dua & Graff, 2019): `metro` and `pm25` (Liang et al., 2015). We chose these datasets in addition as they are of similar size, they are marked as regression tasks and no data points are missing.

Empirically, CGLB seems to better estimate the quadratic term, whereas ACGP is faster to identify the log-determinant. Figure 2 shows a typical example. Note that, for the quadratic form, the upper bounds tend to be less tight than the lower bounds. Generally, there is no clear winner; sometimes ACGP estimates both quantities faster and sometimes CGLB. For other results, see the appendix.

The reason why CGLB has more difficulties to approximate the log-determinant is that the bound involves $\text{trace}[\boldsymbol{K} - \boldsymbol{Q}]$ where $\boldsymbol{Q}$ is a low rank approximation to $\boldsymbol{K}$. If $\boldsymbol{K}_{\text{ff}}$ is of high rank, the gap in the trace can be large. For CGLB the time to compute the bounds is dominated by the pivoted Cholesky decomposition to select the inducing inputs. This overhead becomes irrelevant for the following hyper-parameter tuning experiments, since the selection is computed only once in the beginning. One conclusion from these experiments is to keep in mind that when high precision is required, simply computing the exact solution can be a hard-to-beat baseline.

## 4.2 Application in hyper-parameter tuning

We repeat the hyper-parameter tuning experiments performed by Artemev et al. (2021) using the same set-up, see Appendix A for details. We use the same kernel function, a Matérn $\frac{3}{2}$, and the same optimizer: L-BFGS-B (Liu & Nocedal, 1989) with SCIPY (Virtanen et al., 2020) default parameters. Artemev et al. (2021) report their best results using $M = 2048$ inducing inputs. For reference, we also compare against Sparse Variational Gaussian process regression (SGPR) by Titsias (2009) initialized with the same 512, 1024 and 2048 inducing inputs as CGLB. We use root mean square error (RMSE), negative log predictive density (NLPD) and exact, marginal log-likelihood on the training set, $\log p(\boldsymbol{y})$, as performance metrics. The results for all experiments discussed in this section can be found in Appendix B.1. Here, we will focus on the behavior of each method during training.

A possible application of ACGP is that an optimizer can decide how precise function evaluations need to be. To explore this possibility, we successively decrease the "relative change in function

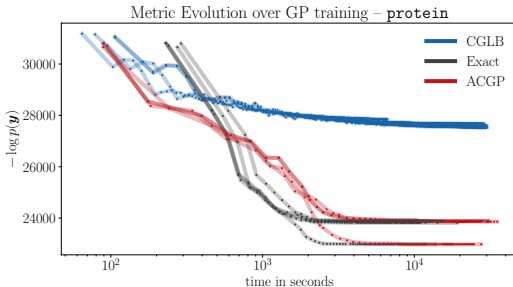
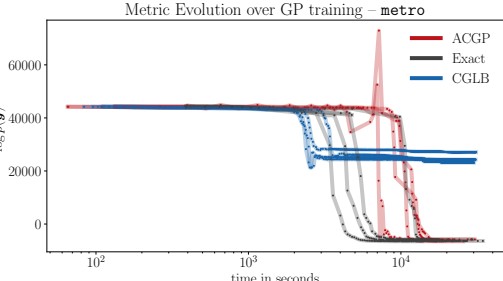

(a) `protein` dataset. The iteratively increasing precision may allow ACGP to reach better solutions faster than exact inference at the price of later convergence.

(b) `metro` dataset. Function evaluations with CGLB are generally the fastest at the cost of plateauing at higher objective function values.

Figure 3: Typical examples of the evolution of the exact log marginal likelihood $p(\boldsymbol{y})$ while optimizing hyper-parameters. See Appendix B.1 for additional plots for all datasets, as well as for SVGP runs.

value" (`ftol`) convergence criterion of L-BFGS-B as $(2/3)^{\text{restart}+1}$ and set this as value for $r$. *With this choice, ACGP does not have any more free parameters than a standard optimizer.* The block size is a problem independent parameter and it is set to the same value as in Section 4.1.

We explore two different computing environments. For datasets smaller than 20 000 data points, we ran our experiments on a single GPU. The results can be summarized in one paragraph: all methods converge the latest after two minutes. The time difference between methods is less than twenty seconds. Exact Gaussian process regression is fastest, more often than not. The results can be found in Appendix B.1. We conclude that in an environment with significantly more processing resources than memory, approximation may just cause overhead.

For datasets larger than 20 000 datapoints, our setup differs from Artemev et al. (2021) in that we use only CPUs on machines where the kernel matrix still fits fully into memory. On all datasets, ACGP is essentially exhibiting the same optimization behavior as the exact Gaussian process regressor, just stretched out. ACGP can provide results faster than exact optimization but may be slower in convergence as Figure 3a shows for the `protein` dataset. This observation is as expected. However, approximation can also hinder fast convergence as Figure 3b reveals on for the `metro` dataset. CGLB benefits from caching the chosen inducing inputs and reusing the solution from the last solved linear equation system. The algorithm is faster, though it often plateaus at higher objective function values. The results for `kin40k` are similar to `protein` and the results for `pm25` are similar to `metro`. These results and the evolution of the root mean square error over time can be found in the appendix. Again, when the available memory permits, the exact computation is a hard-to-beat baseline. However, the Cholesky as a standard numerical routine has been engineered over decades, whereas for the implementations of CGLB and ACGP there is opportunity for improvement.

## 5 Conclusions

In this paper we have revisited the use of Cholesky decompositions in Gaussian process regression. We have shown that the Cholesky decomposition almost computes expected lower and upper bounds on the marginal log-likelihood associated with GP regression. With only small modifications to this classic matrix decomposition, we can use these bounds to stop the decomposition before all observations have been processed. This has the practical benefit that the kernel matrix $\boldsymbol{K}$ does not have to computed prior to performing the decomposition, but can rather be computed on-the-fly.

Empirical results indicate that the approach carries significant promise, but no clear winner can be crowned from our experiments. In general, we find that exact GP inference leads to better behaved optimization than approximations such as CGLB and inducing point methods, and that a well-optimized Cholesky implementation is surprisingly competitive in terms of performance. An advantage of our approach is that it is essentially parameter-free. The user has to specify a requested numerical accuracy and the computational demands will be scaled accordingly. Finally, we note that ACGP is complementary to much existing work, and should be seen as an addition to the GP toolbox, rather than a substitute for all existing tools.

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
