# OpenReview forum: "Adaptive Cholesky Gaussian Processes"
_NeurIPS.cc/2022/Conference — NeurIPS 2022 Submitted_

### Official Review · Reviewer_xgeQ · 2022-07-08

**Rating:** 6
**Confidence:** 3
**Soundness:** 2 fair
**Presentation:** 4 excellent
**Contribution:** 3 good

**Summary:**

The paper provides a stopping criterion for random data subsampling (without replacement) for Gaussian process inference by computing upper and lower bounds to the final log marginal likelihood.
These bounds are shown to hold on expectation, which I understand to be taken over the randomness of the subsampling order.
Importantly, the bounds can be computed as a by-product from computing the Cholesky decomposition of the kernel matrix that would need to be computed anyway by the standard exact inference algorithm. That means that one can determine the random subset without computational overhead (at least in terms of asymptotic order of growth) and on stopping one already ends up with the Cholesky decomposition of the kernel matrix with respect to the data subset.
In particular, the authors use the standard decomposition of the log marginal likelihood into a quadratic and log determinant term, and develop bounds for both that can be computed efficiently from the partial Cholesky decompositions.
In contrast to previous methods for large scale GP inference, e.g., those that construct representative subsets of inputs or pseudo-inputs, the presented methods can be computed without seeing all the input data. This makes it appealing for very large scale datasets where small samples can be sufficient to approximately determine the posterior.
An experimental evaluation of the idea is provided that shows that the two individual bound pairs (log determinant and quadratic) can perform rather differently compared to a related approach that uses conjugate gradients. In particular the bound on the log determinant seems to be more efficient.


**Questions:**

What is the reason for the peak in the log marginal likelihood in the second hyperparameter optimisation experiment? I might not understand the experimental setup here. Should these curves not be monotonically decreasing?


**Limitations:**

The authors strive to honestly discuss the limitations of the work. Two things that should probably be considered in more detail:
1. While the computation of the stopping criterion does not affect the asymptotic order of growth of the computational complexity there seems to be some notable overhead that leads the method to be computationally outperformed by exact inference in some cases. This overhead should be discussed in more detail.
2. Assumption 1 is merely claimed to “appear empirically true”. What is the evidence for that? More discussion or experiments would be useful.


**Strengths And Weaknesses:**

Strengths:
* The idea is to my knowledge novel and appealing because of its principal independence of the overall sample size and its mathematical aesthetics as a function of the partial Cholesky decompositions. One could imagine these bounds to be useful in various contexts.
* Also the paper is very well written and presents technically challenging ideas in an accessible manner in a small amount of space.

Weaknesses:
* The bounds are only shown to hold on expectation and, while the derivation of a full probabilistic guarantee might be out-of-scope for this work, there seem to also be no experiments that test in a representative range of settings the probability with which the bounds hold in practice.
* Generally, the tested settings seem rather limited with only one dataset considered in the main text and two more in the appendix. Given that the theory is still in early stages a little more empirical testing would be useful.
In particular, since the idea seems so appealing, one would expect to see some synthetic settings that demonstrate that the proposed method can outperform existing approximate and exact inference methods by a wide margin. However, such settings are not developed.
* Also for the datasets that are considered (at least in the main text), per dataset we only see the performance of the bounds of one component of the log marginal likelihood. While the analysis on this level of detail is certainly interesting, the overall performance would depend on the interplay of both bounds per dataset.

---

> ### Author Response · Authors · 2022-08-02
> **Author response**
>
> Thank you for your careful review and great feedback. We are happy that you see the potential of our work.
>
> **Weaknesses**
>
> *Point 1:*
> We are addressing this point with our experiments in Section 4.1, for which more results can be found in Appendix B.3. We conjecture that our bounds can be improved since we did not observe any failures.
>
> *Point 2:*
> First, kindly note that, in total, we have investigated eight datasets, cf. Table 1 in Appendix A.
>
> Also, thank you for acknowledging (in Limitations) that we aim to show the limitations of our work. We believe that these limitations are more important to show because synthetic settings where our method outperforms all other approaches are easy (almost trivial) to devise. You already gave one example yourself (very large scale datasets where small samples can be sufficient to approximately determine the posterior). Further examples where ACGP shines:
> 1) Consider a synthetic, densely-sampled and extremely large dataset where even $O(N)$ operations are intractable. Taken to the extreme: the dataset consists of a single datapoint, copied $N$ times. In this case, our algorithm will stop after processing $M=m=10240$ datapoints.
> 2) Consider an expensive kernel function, for example string or graph kernels. Again taken to the extreme: a kernel that blocks for $t$ seconds. Making $t$ large enough, our approach can beat any competitor just by saving kernel evaluations.
> We will point out these advantages more clearly.
>
> *Point 3:*
> We were aiming to show a more detailed picture; however, we agree that seeing the bounds on the actual log marginal likelihood is also important. We will add such plots to the revised paper, which will be uploaded in a couple of days.
>
> **Questions**
>
> Thank you for pointing us to a peculiarity. It appears we inherited a bug from the CGLB optimization procedure (https://github.com/awav/CGLB/blob/main/cglb/backend/pytorch/interface.py#L327). When optimization restarts, it does so from the last assigned parameter values. If the last step was a failed/crashed linesearch, the objective function may increase. We are rerunning our experiments making sure that optimization restarts only from accepted parameter values. At this point, it appears that fixing this mistake is to our advantage since it was mostly ACGP suffering from bad restarts. In the RMSE plot, the spike has disappeared. The $-\log p(y)$ computations are still running, and we will update all plots in the revised paper as soon as the computations are done.
>
>
> **Limitations**
>
> *Point 1:*
> Just above Section 2, we state that our stopping criterion induces an $O(M)$ overhead. For the experiments in Section 4.1 (and Appendix B.3), the overhead is, in fact, less than it appears in the manuscript: we forgot to add the time to construct the kernel matrix to the timings of the exact GP, which almost doubles the compute time for this baseline. This makes the compute time of the exact GP much more comparable to that of ACGP, showing that the actual overhead of ACGP is little. We will update all plots in the revised paper.
>
> In Section 4.2, exact inference outperforms as well our strongest competitor: CGLB. A surprising insight from our experiments is that, although exact inference is expensive, the costs can amortize over the course of optimization. We believe this is an important message for the GP community.
>
> *Point 2:*
> For the experiments in Section 4.1, we checked if the empirical quantities satisfied the assumption, which we can confirm. We will add this information below the statement of the assumption.
>
> The assumption essentially says that the mean prediction error decreases with more data. We suspect that one can construct kernel functions for which this is not the case (hence, we need to assume), but a model that becomes worse with more data is unlikely to be an interesting model anyway.

---

### Official Review · Reviewer_rfLj · 2022-07-11

**Rating:** 4
**Confidence:** 3
**Soundness:** 2 fair
**Presentation:** 2 fair
**Contribution:** 1 poor

**Summary:**

This paper proposed a generic marginal likelihood approximation in Gaussian process models by using a subset of the data. A logged marginal likelihood is written as  a summation of recursive logged predictive densities. Although a predictive density is analytically calculated, the computation associated with a covariance matrix becomes expensive easily with a number of data points.
the data is divided into two folders, a processed A (size M) and a remaining B (size N-M).  A logged marginal likelihood has two terms, a determinant and a quadrative term. Instead of taking a recursive sum, a logged marginal likelihood is bounded using a subset of the data (say, s-samples in A folder) and the logged marginal likelihood is approximated by the median of a lower and an upper bounds.

The paper presented the bound and proof and numerical simulation study shows that a tight bound can be achieved using some subset data, s.


**Questions:**

Please see Strengths and Weakness.

**Limitations:**

yes.

**Strengths And Weaknesses:**

The marginal likelihood is a model evidence that it is an important estimation, particularly for model comparison. Any reliable evidence approximation will receive lots of attention.

There are some uncertainties and unclearness about the method.

a)	Bounds  (both  a determinant and a quadratic term) were formed by for by a posterior for s data in the A folder and a weighted/adjusted prediction for the remining data in A (M-s). Such argument makes sense for one prediction, p(x_{s+1}|x_{s}) and I do not get why this is a reasonable formation for a summation of recursive predictions.
b)	The approximation will be sensitive to M (size of the A folder) and s (subset used for a posterior). It is clear that  as s approaches N, we will get the exact evidence. I feel the paper did not investigate enough to present the approximation quality with s and M both theoretically and numerically.
c)	I don’t think authors did not provide any guide on how to choose s and M. In the simulation study, only M values are shown, no s values.
d)	Could authors comment on the behavior of bounds with s/M for different model dimensions?
e)	From the practical point of view, s is likely to be chosen with a tight bound using some threshold. When we compare models, it is crucial that we get the same comparison result using evidence approximations. Could authors comment on whether your method will be reliable in this context?

---

> ### Author Response · Authors · 2022-08-02
> **Author response**
>
> Thank you for your review and the discussion points you raise.
>
> a) Our proof sketch in Appendix D.3 may help to clarify this question. Particularly the paragraph from line 160 addresses your point: the remaining posterior predictions are correlated, which is why we look at bounds that remove this correlation.
>
> b) We investigate this question in Section 4.1, for which the complete results can be found in Appendix B.2.
>
> c) Note that $M$ is not a choice but an adaptive stopping time that depends on dataset and kernel. We describe how it is chosen in Section 3.2. Also $s$ is not a choice but defined via $M$ and the sample set size $m$ as $s:=M-m$. Indeed an open question is how large $m$ should be. It must be sufficiently large to obtain reliable bounds. With $m \approx 10000$, our bounds hold in all experiments. Just above Section 3.5, we describe that we could obtain high-probability guarantees if we could define $s$ as a stopping time, meaning independent of $M$. However, it is unclear how one would define such a stopping time. With this submission, we address the question: when can we stop computing? The choice of $s$ is the answer to the question: when can we start to consider stopping? One purpose of this submission is to get the community interested in answering this question together.
>
> d) The dimensionality of the dataset does not enter at any point in our equations. Please see table 1 in Appendix A for sizes and dimensionalities of the datasets that we considered in the experiments and Appendix B.3 for the behavior of the bounds in these datasets.
>
> e) In general, reliably identifying one model as better than another is only possible if the approximation bounds for both models do not overlap; that is, the lower bound for one model is better than the upper bound for the other. In the probabilistic case, for example with stochastic variational inference, bounds only hold with a certain probability, and how reliable and repeatable these estimates are depends on the probabilistic parameters, for example the batch size. As pointed out before, we observed that our bounds always hold when choosing the sample set size $m$ on the order of 10000, and in this case, it should be safe to use ACGP for model comparison.

---

### Official Review · Reviewer_shrn · 2022-07-11

**Rating:** 3
**Confidence:** 4
**Soundness:** 2 fair
**Presentation:** 2 fair
**Contribution:** 2 fair

**Summary:**

This paper proposes speeding up Gaussian Process inference by estimating the log marginal likelihood with a subset of the data inexpensively chosen on the fly.

The core idea is to bound the full log marginal likelihood by two quantities $L$ and $U$ that may be estimated inexpensively with a small subset of the data grown incrementally, to stop growing the subset when the error $|U-L|$ is small enough, and to use as estimate for the log marginal likelihood $(U+L)/2$.

**Questions:**

Please address the concerns in bold above. Additionally, what are the fundamental differences between what you are trying to achieve and inducing point selection?

**Limitations:**

Yes.

**Strengths And Weaknesses:**

While the core idea above is interesting, the implementation of this idea by this paper has a few fundamental limitations, both conceptual and theoretical.


## Conceptual Limitations

**The first $M$ points might not be representative of the input space or even the full dataset**

The fundamental idea underpinning the stopping strategy proposed in this paper is that, if we estimate the log marginal likelihood of $M$ data points selected sequentially, and we realize that the last $s$ points we added were somehow redundant in the estimation of the log marginal likelihood of the $M$ points, then we can conclude that the first $(M-s)$ points selected are good enough to compute the log marginal likelihood of ***any*** $n \gg M$ points.

This is clearly not the case!

**Counter-Example:** As a simple counter-example, consider the following toy 1D regression problem: $x_i = i\frac{2\pi}{n}, ~ i \in [1, n]$ with $n=1000000$, and $y_i = \sin(x_i)+\epsilon_i$ where $\epsilon_i$ are i.i.d. Gaussian noise terms with standard deviation $1/100$. Basically, a noisy sine function.

Clearly, inputs $x_i$ are sorted and, no matter the interval $[0, \alpha]$ you consider, you do not need nearly as much as $1000000\frac{\alpha}{2\pi}$ points to represent the sine function! You don't even need $100\frac{\alpha}{2\pi}$ points!

For most $M$ you'll take, you'll find that points $(x_{M-s}, \dots, x_M)$ are redundant (too many points relative to how fast the function is changing) given $(x_1, \dots, x_{M-s})$.

Yet, no matter $x_M \in [0, 2\pi)$, knowing $(y_1, x_1), \dots, (y_M, x_M)$ is clearly not sufficient to estimate the full log marginal likelihood for the simple reason that you have no clue what the latent function can possibly look like beyond $x_M$.

Is the latent function smooth, is it piecewise smooth, does it undergo changepoints after $x_M$, etc. All these scenarios would yield drastically different full marginal likelihood and you simply cannot tell them apart simply from observing $(y_1, x_1), \dots, (y_M, x_M)$.

In general, because points $(y_{M-s}, x_{M-s}), \dots, (y_M, x_M)$ are redundant relative to points $(y_1, x_1), \dots, (y_{M-s}, x_{M-s})$ does not mean that $(y_1, x_1), \dots, (y_{M-s}, x_{M-s})$ are representative of the latent function on the **whole input domain**, which is what you need.

**The main takeaway here is that the order in which you consider your inputs $x_i$ matters! At the very least, your dataset needs to be shuffled, you need to revise your algorithm to make sure it is not exposed to the peculiarities of a specific permutation, and you need to provide some analysis on the effect of this random shuffling on overall performance.**


## Theoretical Limitations

The paper also makes a few theoretical mistakes.


**Data Points Are Not i.i.d.; Theorem 2 is Questionable.**

First, it is said a couple of times that data $(y_i, x_i)$ are assumed **i.i.d.** (e.g. Page 4, Line 125, and Theorem 2). That's incorrect and (obviously) never the case in Gaussian Process modeling. If that was the case, the Gram matrix $K$ would be diagonal and you would not have any issue estimating the marginal likelihood.

$(y_i, x_i)$ are usually assumed **i.i.d. conditional on the latent function**! In other words, it is the noise terms affecting observations that are assumed to be i.i.d., not the observations.

It would appear that the main result, Theorem 2, relies on this assumption.


**The Upper Bound $\mathcal{U}_D$ is questionable**

Note that the log determinant of a positive definite matrix is always smaller or equal to the log determinant of its diagonal.

Hint: Relate the log-det of a psd matrix to the entropy of a multivariate Gaussian, note that the entropy of a multivariate distribution is the sum of the entropies of its marginals and the entropy of its copula, and note that the entropy of a copula is always non-positive (the independence copula is maximum-entropy and has entropy $0$).

Hence, in Equation (6), the term $(N-s)\mu_D$ should really be the sum of the posterior variances of $y_{s:N}$. No need for the incorrect i.i.d. assumption on $(y_i, x_i)$.


**The Lower Bound $\mathcal{L}_D$ is questionable**

Fundamentally, is there really any hope to lower-bound the log-det of the conditional covariance matrix? Depending on how correlated any two unobserved outputs are unconditionally, this conditional covariance matrix can have as low a determinant as possible, irrespective of any observations we have made.

If any two unobserved outputs are perfectly correlated, which depends solely on the kernel, then the log-det of the conditional covariance matrix will be $-\infty$, no matter what outputs $y_{:s}$ were observed!

---

> ### Author Response · Authors · 2022-08-02
> **Author response**
>
> Thank you for your very detailed review and the time and effort you put into it. Also, thank you for pointing out a mathematical subtlety that people in the GP community might find confusing; we are applying a frequentist's perspective to approximate numerical quantities for a Bayesian---but no longer being Bayesian. In the Bayesian perspective, it indeed makes no sense to assume that inputs and targets are jointly independent. In the frequentist literature, though, this assumption is common and also sensible, because all probabilities are conditioned implicitly on the true underlying function (which may not even be part of the RKHS). Since that function is a deterministic quantity, it is dropped quietly from the conditioned variables. We will add this clarification to our introduction.
>
> Hoping that we convinced you that the i.i.d. assumption is admissible, please note that your counter-example violates it. Via the i.i.d. assumption, we make transparent that our algorithm can fail when applied to datasets with a peculiar permutation or ordering (time-series being an example). We sacrifice guarding against these cases in exchange for independence of $N$---this is where the speed-up is coming from. We provide an analysis with the proof of Theorems 2 and 3. As you suggested, in our experiments, we indeed shuffle our datasets and report results corresponding to five independent permutations.
>
> You are correct that $y_{s:N}$ is indeed a deterministic upper bound to the remaining determinant. The critical observation is that we do not bound this quantity but the expectation of that term conditioned on the first $s$ inputs. Regarding the lower bound: that there is hope has definitely been shown in the peer-reviewed CGLB paper (Artemev et al., 2021). The proofs for our claims may appear somewhat lengthy, but note that we provide a more accessible proof sketch in Appendix D.

---

> > ### Comment · Reviewer_shrn · 2022-08-05
> > **Re: Author response**
> >
> > Thanks for your response.
> >
> > **Re: The i.i.d. assumption is admissible**.
> >
> > Being frequentist here isn't a matter of choice. If you work conditional on $f$ or assume $f$ is deterministic, then $(y_i)$ would be independent, $K$ would be diagonal, and the marginal likelihood would be easy to evaluate. If you are concerned with speeding up the evaluation of the log-marginal likelihood, then you cannot possibly claim that $(y_i, x_i)$ are i.i.d. unconditionally.
> >
> > Simply put: $(y_i, x_i)$ i.i.d => $(y_i)$ i.i.d. => $(y_i)$ independent => $(y_i)$ decorrelated => $K$ diagonal => no marginal likelihood computation problem. (=> : implies).
> >
> > I think you are confusing the randomness inherent to regression models with the randomness you need for this approach. What you want here is a (sample) randomness that guards against any specific/peculiar order in which samples are sequentially picked.
> >
> > Rigorously, you want to introduce a random (and uniform) permutation $(\pi_i)$ of $(1, \dots, N)$, and apply your algorithm to $(y_{\pi_i}, x_{\pi_i})$, not to $(y_i, x_i)$. Note that, even then, $(y_{\pi_i}, x_{\pi_i})$ would be identically distributed but not quite independent! Counter-examples are easily constructed.
> >
> > To sum-up the i.i.d. assumption is not admissible. Even if you shuffle your $N$ samples, the identically distributed assumption would become admissible but not the independence assumption.

---

> > > ### Author Response · Authors · 2022-08-08
> > > **Re: The i.i.d. assumption is admissible.**
> > >
> > > Thank you for your follow-up questions. We genuinely think we are in agreement, but that we might be talking past each other.
> > >
> > >
> > > > If you work conditional on $f$ or assume $f$ is deterministic, then $(y_i)$ would be independent, $K$ would be diagonal, […]
> > >
> > > To make an example where $K$ is not diagonal, but the $y$’s are independent:
> > > Consider for instance $N$ $x$’s and $y$’s sampled from a 1D zero-mean, unit-variance Gaussian. These $x$’s and $y$’s will be i.i.d., but for a choice of kernel of, say, a squared exponential with length scale 1, the kernel matrix $K$ will not be diagonal.
> > >
> > > Thus, even if the $y$’s are independent, $K$ may not be diagonal. We hope we can agree on this example.
> > >
> > >
> > > > If you are concerned with speeding up the evaluation of the log-marginal likelihood, then you cannot possibly claim that $(x_i, y_i)$ are i.i.d. unconditionally.
> > >
> > > We very much agree with this! As we pointed out in our response to your initial review, we condition implicitly on the true underlying function $f$.
> > >
> > > Please note that the i.i.d. assumption does not carry over to the elements of the kernel matrix. Writing $y_n=f(x_n)+\epsilon_n$, one can see, for example for the quadratic form, that analyzing the expectation of
> > >
> > > $$\vec y^\top K^{-1}\vec y=[f(x_1)+\epsilon_1, \ldots, f(x_N)+\epsilon_N] \begin{bmatrix}k(x_1,x_1)+\sigma^2 & k(x_1, x_2) & \ldots\\\\ \vdots & \ddots \end{bmatrix}^{-1}[f(x_1)+\epsilon_1, \ldots, f(x_N)+\epsilon_N]^\top$$
> > >
> > > is non-trivial, due to the application of the kernel function. Coming back to the earlier point: independent $x$’s can have non-zero kernel function values giving rise to non-zero off-diagonal entries.
> > >
> > >
> > >
> > > > Simply put: $(y_i, x_i)$ i.i.d => $(y_i)$ i.i.d. => $(y_i)$ independent => $(y_i)$ decorrelated => $K$ diagonal => no marginal likelihood computation problem. (=> : implies).
> > >
> > > We agree with all the implications except for the next to last one (=> $K$ diagonal) for the reasons discussed in the counter-example above. That the dataset entries (conditioned on the true function) are i.i.d., does not enforce a diagonal kernel matrix.
> > >
> > > To reiterate, even independent $x$’s can have non-zero kernel function values, since two independent points can (and probably will) have non-zero Euclidean distance.
> > >
> > >
> > > > I think you are confusing the randomness inherent to regression models with the randomness you need for this approach. What you want here is a (sample) randomness that guards against any specific/peculiar order in which samples are sequentially picked.
> > >
> > > We are not sure that we understand what you mean by “model randomness” and “sample randomness”. If by “sample randomness” you are referring to randomness in the dataset, then this is exactly the i.i.d. assumption, the randomness that our algorithm and proofs rely on. In this sense, the i.i.d. assumption is our guard against peculiar dataset ordering.
> > >
> > > The primary goal of our approach is to computationally exploit sample randomness (i.i.d. dataset) beyond prior and likelihood (model randomness) if the former is present. If we misunderstood your comment, please elaborate on randomnesses you are referring to.
> > >
> > >
> > > > Rigorously, you want to introduce a random (and uniform) permutation $(\pi_i)$ of $(1, \dots, N)$, and apply your algorithm to $(y_{\pi_i}, x_{\pi_i})$, not to $(x_i, y_i)$.
> > > > Note that, even then, $(y_{\pi_i}, x_{\pi_i})$ would be identically distributed but not quite independent! Counter-examples are easily constructed.
> > >
> > > Practically speaking, this is exactly what we do in our experiments.
> > >
> > > Theoretically, we agree that introducing a random permutation would be the more rigorous way of proving that our algorithm works for our experiments. However, sampling-without-replacement results are generally hard to obtain. Instead, we rely on the i.i.d. assumption, which is standard practice in the frequentist literature. However, with increasing dataset size, sampling with replacement (which satisfies the i.i.d. assumption) and sampling without (random permutations) become more and more the same. Our experiments (and many before us) validate this empirically.
> > >
> > >
> > > > To sum-up the i.i.d. assumption is not admissible. Even if you shuffle your $N$ samples, the identically distributed assumption would become admissible but not the independence assumption.
> > >
> > > We hope that we have convinced you that the i.i.d. assumption (exemplified with our 1D Gaussian toy example above) does not invalidate the GP model. Assuming data to be i.i.d. is common in the frequentist world, and is how proofs are typically made. Within this framework, our theory and proofs hold. The i.i.d. setting can be established in practice (at least approximately for large datasets) by shuffling the data. This is what our algorithm relies on and why we make sure to shuffle the data in our experiments. We will make this clearer in the paper.

---

> > > > ### Comment · Reviewer_shrn · 2022-08-08
> > > > **Re: Re: The i.i.d. assumption is admissible.**
> > > >
> > > > You might be confused.
> > > >
> > > > Your counter-example isn't one. If $y_i$ are independent then they are decorrelated, and their covariance matrix ought to be diagonal, by definition! If you denote $K$ their covariance matrix, it ought to be diagonal...
> > > >
> > > > In the log-likelihood of a multivariate Gaussian, the matrix $K$  **IS** the covariance matrix of the Gaussian vector. You cannot decouple the nature of $K$ from the distribution of $(y_1, \dots, y_n)$.
> > > >
> > > > Once more, even if you shuffle the dataset $(y_1, \dots, y_n)$, the shuffled versions $y_{\pi_i}$ are identically distributed but **NOT** independent. Hint: if $y_i$ are highly correlated, so will $y_{\pi_i}$.
> > > >
> > > > Randomly sampling with replacement violates much of the narrative in your paper (e.g. sequentially going through the sample, the incremental decomposition of Eq (2), etc.).

---

> > > > > ### Author Response · Authors · 2022-08-09
> > > > > **Re: Re: Re: The i.i.d. assumption is admissible.**
> > > > >
> > > > > Thank you for continuing the discussion.
> > > > >
> > > > > > Your counter-example isn't one. If $y_i$ are independent then they are decorrelated, and their covariance matrix ought to be diagonal, by definition! If you denote $K$ their covariance matrix, it ought to be diagonal...
> > > > > > In the log-likelihood of a multivariate Gaussian, the matrix **IS** the covariance matrix of the Gaussian vector. You cannot decouple the nature of from the distribution of $(y_1, …, y_n)$.
> > > > >
> > > > > You are right that (for our example) the kernel matrix *ought* to be diagonal, but model misspecification happens. The true covariance of the samples can be different from the kernel matrix. Considering again our example: during the course of hyper-parameter tuning, the matrix $K$ would certainly turn more diagonal, but not from the start.
> > > > >
> > > > > The kernel function reflects our believe about how the true function values relate as a function of the inputs. Even though we condition implicitly on this true function, we do not know this relationship. The independence assumption on the dataset is not a contradiction.
> > > > >
> > > > > Put differently, since the kernel function is specified to capture our belief about the true underlying function, it makes sense to choose kernel functions that do not result in diagonal kernel matrices—even in the frequentist setting.
> > > > >
> > > > >
> > > > > > Once more, even if you shuffle the dataset $(y_1, …, y_n)$, the shuffled versions $y_{\pi_i}$ are identically distributed but **NOT** independent. Hint: if $y_i$ are highly correlated, so will $y_{\pi_i}$.
> > > > >
> > > > > Once again, we absolutely agree with you: shuffling the data does not establish strict i.i.d.-ness. However, that this is not an issue is discussed concisely in the following half page article: https://web.ma.utexas.edu/users/parker/sampling/repl.htm
> > > > >
> > > > > Sampling with replacement would establish i.i.d.-ness, and the article discusses why there is only little difference to sampling without (i.e., shuffling).
> > > > >
> > > > >
> > > > > > Randomly sampling with replacement violates much of the narrative in your paper (e.g. sequentially going through the sample, the incremental decomposition of Eq (2), etc.).
> > > > >
> > > > > If we can now agree that the difference between sampling with or without replacement becomes negligible for large datasets, you will maybe acknowledge that it does not violate the narrative of our paper.
> > > > >
> > > > > Thank you for a great discussion on the fundamental principles of the frequentist theory underlying our work. We hope the other reviewers and the AC also benefit from these clarifications.

---

### Author Response · Authors · 2022-08-09
**Message to all reviewers**

Dear reviewers,

Thank you for your feedback on our paper. Having rerun our experiments, we have now uploaded a revised manuscript. In particular, we would like to highlight the changes to Figure 2, where we have corrected a mistake in the timing of the exact GP. The figure now clearly shows that the overhead of ACGP compared to standard Cholesky decomposition is negligible.

Please note that the exact-log-likelihood plots for the hyper-parameter tuning experiments are not yet updated, as these computations take longer. They will be updated for the camera-ready version.

Again, thank you all for your questions and comments. We hope our discussions clarified your questions and added to the soundness of our work. With all concerns addressed, we hope this will reflect in your updated score.

---

### Meta-Review · Area_Chair_6Uxk · 2022-08-23

**Recommendation:** Reject
**Confidence:** Certain

**Metareview:**

This paper proposes some nice ideas on speeding up Gaussian process inference based on approximating the marginal using subsamples. However, several reviewers noted gaps and potentially flaws in the technical details. The reviews as well as detailed replies during the rebuttal period will help the authors prepare a stronger revision, but the work is not airtight and is not ready for publication in its current form

**Award:**

No

---

### Decision · Program_Chairs · 2022-09-14

Reject